# Ratio of Urinary Proteins to Albumin Excretion Shifts Substantially during Progression of the Podocytopathy Alport Syndrome, and Spot Urine Is a Reliable Method to Detect These Pathologic Changes

**DOI:** 10.3390/cells12091333

**Published:** 2023-05-07

**Authors:** Jan Boeckhaus, Lea Mohr, Hassan Dihazi, Burkhard Tönshoff, Lutz T. Weber, Lars Pape, Kay Latta, Henry Fehrenbach, Baerbel Lange-Sperandio, Matthias Kettwig, Hagen Staude, Sabine König, Ulrike John-Kroegel, Jutta Gellermann, Bernd Hoppe, Matthias Galiano, Dieter Haffner, Heidrun Rhode, Oliver Gross

**Affiliations:** 1Clinic for Nephrology and Rheumatology, University Medical Center Goettingen, 37075 Goettingen, Germany; 2Department of Pediatrics I, University Children’s Hospital Heidelberg, 69120 Heidelberg, Germany; 3Pediatric Nephrology, Children’s and Adolescents’ Hospital, Faculty of Medicine and University Hospital Cologne, University of Cologne, 50937 Cologne, Germany; 4Department of Pediatric Kidney, Liver and Metabolic Diseases, Hannover Medical School, 30625 Hannover, Germany; 5Department of Pediatrics II, University Childrens’ Hospital, University of Duisburg-Essen, 45147 Essen, Germany; 6Clementine Kinderhospital Frankfurt, 60316 Frankfurt, Germany; 7Pediatric Nephrology, Children’s Hospital, 87700 Memmingen, Germany; 8Dr. v. Hauner Children’s Hospital, Ludwig Maximilians University, 80337 Munich, Germany; 9Clinic of Pediatrics and Adolescent Medicine, University Medical Center Göttingen, 37075 Göttingen, Germany; 10Pediatric Nephrology, University Children’s Hospital Rostock, 18057 Rostock, Germany; 11University Children’s Hospital Münster, 48149 Münster, Germany; 12Division of Pediatric Nephrology, University Children’s Hospital, 07743 Jena, Germany; 13Pediatric Nephrology, Charité Children’s Hospital, 10117 Berlin, Germany; 14Division of Pediatric Nephrology, Department of Pediatrics, University of Bonn, 53121 Bonn, Germany; 15Department of Pediatrics and Adolescent Medicine, University Hospital, Friedrich-Alexander-University Erlangen, 91054 Erlangen, Germany; 16Department of Internal Medicine I, Cardiology, Angiology, Intensive Medical Care, University Hospital Jena, Am Klinikum 1, 07747 Jena, Germany

**Keywords:** proteinuria, albuminuria, diagnostic marker, renal fibrosis, type IV collagen, Alport syndrome, podocytopathies, hereditary kidney diseases, podocytes, glomerulus, mechanical stretch

## Abstract

The urinary albumin- and protein-to-creatinine ratios (UACR and UPCR, respectively) are key endpoints in most clinical trials assessing risk of progression of chronic kidney disease (CKD). For the first time, the current study compares the UACR versus the UPCR head-to-head at early stages of CKD, taking use of the hereditary podocytopathy Alport syndrome (AS) as a model disease for any CKD. Urine samples originated from the prospective randomized, controlled EARLY PRO-TECT Alport trial (NCT01485978). Urine samples from 47 children with confirmed diagnoses of AS at very early stages of CKD were divided according to the current stage of AS: stage 0 (UACR < 30 mg/g), stage 1 (30–300 mg/g) or stage 2 (>300 mg/g). The range of estimated glomerular filtration rate was 75–187.6 mL/min. The mean age was 10.4 ± 4.5 years. In children at stage 0, proteinuria in spot urine, confirmed in 24 h urine, was almost ten times higher than albuminuria (106.4 ± 42.2 vs. 12.5 ± 9.7; *p* < 0.05); it was “only” about three times higher in stage 1 (328.5 ± 210.1 vs. 132.3 ± 80.5; *p* < 0.05) and almost equal in stage 2 (1481.9 ± 983.4 vs. 1109.7 ± 873.6; *p* = 0.36). In 17 children, UACRs and UPCRs were measured simultaneously in 24 h urine and spot urine in the same study visit. Interestingly, the UACR (and UPCR) in 24 h urine vs. in spot urine varied by less than 10% (266.8 ± 426.4 vs. 291.2 ± 530.2). In conclusion, our study provides the first evidence that in patients with normal glomerular filtration rate (GFR) and low amounts of albuminuria, especially in children with podocytopathies such as AS, measuring the UACR and UPCR in spot urine is a reliable and convenient alternative to 24 h urine collection. Our study advocates both the UACR and the UPCR as relevant diagnostic biomarkers in future clinical trials in children with glomerular diseases because the UPCR seems to be a very significant parameter at very early stages of podocytopathies. The German Federal Ministry of Education and Research funded this trial (01KG1104).

## 1. Introduction

The urinary albumin-to-creatinine ratio (UACR) is a very common marker for risk of progression of diabetic and nondiabetic chronic kidney disease (CKD) [1,2]. In addition to the UACR, the urinary protein-to-creatinine ratio (UPCR) is used to assess kidney function. Both the UACR and the UPCR are very important primary and key secondary endpoints in most clinical trials in patients with CKD.

There are no randomized controlled trials of direct comparison between proteinuria and albuminuria. Therefore, any study assessing the association between proteinuria and albuminuria in CKD faces the problem of indirect comparisons [3].

The current study uses the hereditary type IV collagen-related nephropathy Alport syndrome (AS) as a model disease for podocytopathies in order to assess the relationship between the UACR and the UPCR, which, to our knowledge, has never been analyzed before at very early stages of disease in any podocytopathy. AS is the most common monogenetic kidney disease. It is caused by variants in the genes *COL4A3*, *COL4A4* and *COL4A5*, which encode the α3, α4 and α5 chains of type IV collagen [4,5,6]. Without treatment, AS leads to end-stage kidney disease (ESKD) early in life and is also characterized by typical ocular abnormalities and sensorineural hearing loss [7].

Type IV collagen is an essential component of the basement membranes in the kidney, cochlea and eyes. The pathological hallmark of AS is the disruption of the type IV collagen structure, which leads to dysfunction of the podocytes, with thinning, splitting and thickening of the glomerular basement membrane of the kidney and a well-defined clinical course starting with microscopic hematuria, microalbuminuria and proteinuria and finally progressing to renal fibrosis and ESKD [5]. Though the majority of patients with AS have X-linked (*COL4A5*) inheritance, up to 30% have autosomal recessive or dominant (*COL4A3* or *COL4A4*) inheritance [4]. Heterozygous patients with autosomal dominant AS and female heterozygous X-linked patients with AS showed large variability in the clinical course, with low to intermediate lifetime risk of ESKD (1% to 40%). Prognosis in heterozygous patients largely depends on random X-inactivation and environmental factors such as lifestyle, including smoking, high blood pressure, obesity and an unhealthy diet with high salt and animal meat intake [8]. Due to this large variability in clinical outcome, predictors, such as the UACR and the UPCR, for rapid loss of kidney function are needed.

In a mouse model of AS, doubling of the lifespan until ESKD was observed when therapy with an angiotensin-converting enzyme inhibitor (ACEi) was started before onset of proteinuria [9]. Registry data have shown a similar beneficial effect in humans; progress of renal manifestation can be delayed by treatment with ACEi [10]. To clarify whether an earlier start with ACEi prior to onset of proteinuria (in CKD stage 0 or 1) is safe and effective, the EARLY PRO-TECT Alport trial (NCT01485978) was conducted [11]. Children with AS, aged between 2 and 18 years and with normal glomerular filtration rates, were included in this randomized, placebo-controlled, double-blind trial. The primary efficacy endpoint was disease progression, defined as tripling (in stage 0, if albuminuria at screening was below 30 mg of albumin per gCrea) or doubling (in stage 1, if microalbuminuria at screening was between 30 and <300 mg of albumin per gCrea), or, in any progress to stage 2, defined as albuminuria >300mg of albumin/gCrea. The primary efficacy endpoint was assessed with urine collections in 6-month intervals over a treatment period of 3 to 6 years. Primary safety-endpoint “adverse events” and additional clinical data were assessed with electronic case report forms (eCRFs). As a result, early initiation of ramipril therapy showed no safety concerns compared with a placebo and decreased risk of disease progression by almost half (although without statistical significance) [11].

For market approval of new nephroprotective drugs, the U.S. Food and Drug Administration (FDA) may accept relevant changes in the UACR as primary endpoints and surrogate parameters for CKD progression. Similarly, the phase 3 EARLY PRO-TECT Alport clinical trial measured proteinuria and albuminuria in order to assess the primary endpoints in spontaneous urine and in—if possible with the child—24 h urine collection every six months to monitor disease progression.

In the present study, we hypothesized that the UACR-to-UPCR ratio in children with Alport syndrome at early stages of CKD changes during disease progression because of increasing damage to the glomerular filtration unit. In addition, the purpose of this study is to evaluate variances in UACRs and UPCRs collected from spontaneous urine versus 24 h urine in order to enrich statistical power calculations in future clinical trials aiming for market approval of new medications for patients at early stages of CKD.

## 2. Materials and Methods

### 2.1. Clinical Data and Sampling of Urine

The demographical and clinical data were collected as part of the EARLY PRO-TECT Alport trial; the patient cohort was described previously [11,12,13]. The trial was registered at www.ClinicalTrials.gov (NCT01485978); the EudraCT number is 2010-024300-10.

In brief, children with definite diagnoses of AS, aged between two and 18 years and with normal glomerular filtration rates were included in the trial. Only “classical” forms of Alport syndrome were included: X-linked AS in males, or homozygous or compound-heterozygous autosomal AS in children (both sexes). All patients were characterized genetically; however, in order to qualify for this study, the diagnosis of AS could also be confirmed with kidney biopsy of the child or a close relative. Kidney biopsy had to show classical alternations of the glomerular basement membrane, which are thinning, splitting and thickening of the GBM.

Twenty children were included in the randomized, placebo-controlled, double-blind trial, and forty-two became an open-arm control. For ethical reasons, the EARLY PRO-TECT Alport trial included also an open-arm control for patients whose parents refused randomization in their child (because most families had a positive family history of early ESKD in a close relative). Ramipril (or the placebo) was slowly uptitrated from 1 mg per square meter to the maximum tolerated dose (a maximum of 6 mg per square meter). The dosage was adjusted according to weight and growth during this study, in six-month intervals. The number of adverse events related to low blood pressure or hypotension did not differ between the placebo and Ramipril groups. During the trial, proteinuria was measured in spontaneous urine at screening, at baseline and every six months until the end of this study at month 36 to 72 or at unblinding because of disease progression. After unblinding, the study protocol allowed open-label treating of the children until the end of this study. If possible, examination of 24 h urine was performed.

Disease progress during the trial was defined as either doubling or tripling of albuminuria depending on the stage of the disease at the baseline. The results of the urine samples, vital signs, blood pressure, medications, side effects of medication, previous medical history, family history, eye and ear examinations and adverse and severe adverse events were documented in the trial’s electronic case report forms (eCRFs). All data were pseudonymized. According to the German Medicines Act, the trial was approved by all ethics committees and the Federal Institute for Drugs and Medical Devices (BfArM). Written informed consent was obtained from all legal representatives, as was assent from all patients ≥ 6 years of age.

Spot urine and 24 h urine were collected using standard operating procedures (SOPs) and electronic case report forms (eCRFs) as part of the phase 3 clinical trial and according to the published trial protocol. Each time of start and end of collection of 24 h urine was documented, as well as the total amount of urine. Collection started with the second morning urine (the first morning urine was discarded) and ended with the first morning urine of the next day. Spot urine as midstream urine was collected in the trial center using the second or third morning urine (depending on arrival at the trial site). Urine analysis was performed at the laboratories of the local trial sites. The eCRF automatic program checked the results for plausibility (of creatinine levels and albumin levels in the urine) and for primary efficacy endpoints (doubling or tripling of albuminuria). In addition, progress of disease for each trial participant was checked by the coordinating principal investigator every two to three months.

In this analysis, we used the albuminuria and proteinuria data from the trial sites, which reported both proteinuria and albuminuria in mg/gCrea. When multiple data were available at different time points, those of simultaneous spontaneous and 24 h urine samples were selected. The urinary albumin-to-protein ratio was calculated by dividing the UACR by the UPCR.

### 2.2. Sodium Dodecyl Sulfate Polyacrylamide Gel Electrophoresis (SDS-PAGE) of Urinary Proteins

For urine protein pattern analysis, samples from 15 different children with AS were collected and analyzed using SDS-PAGE according to Laemmli’s method [14]. Each patient urine sample of a 500 µL volume was subjected to a chloroform–methanol precipitation according to Wessel and Flügge [15]. The obtained protein pellet was dissolved in NuPAGE LDS sample buffer (Thermo Fisher, Waltham, MA, USA). The NuPage sample-reducing agent was added according to the manufacturer’s instructions.

In order to obtain an overview of the albumin amount in urine samples, an albumin standard was prepared with 1.5 µL bovine serum albumin (BSA) and 2 mg/mL Bio Rad (Thermo Fisher, Waltham, MA, USA). Next, the samples and the standard were denatured by incubation at 70 °C for 10 min. The samples were centrifuged briefly and redissolved. Nu-PAGE was performed using 4–12% Bis-Tris gel and Nu-PAGE MES SDS running buffer (Thermo Fisher, Waltham, MA, USA). NuPage Antioxidant (Thermo Fisher, Waltham, MA, USA) was added to the running buffer.

To mark molecular protein weights, a protein ladder (QuadColor Protein Marker, 4.6 kDa to 300 kDa (Lonza, Basel, Switzerland)) was used. First, polyacrylamide electrophoresis was carried out at 85 volts for 15 min and then at 100 volts for 1 h. This electrophoresis was followed by overnight staining with READYBLUE ProteinGel Stain (Sigma Aldrich, St. Louis, MI, USA) and then washing with water.

The gel was scanned on a white screen using an Epi-White molecular imager (Bio Rad, Hercules, CA, USA), and documentation was performed using Image Lab software (version 6.0, Bio Rad, Hercules, CA, USA).

### 2.3. Statistical Methods

For this analysis of UACR vs. UPCR, statistical comparisons were not formally powered or prespecified. Data collection and descriptive statistics were performed with Microsoft Excel (version 16.6 for MacOS, Microsoft, Redmond, WA, USA). An unpaired t-test was used to compare mean values. A probability (*p*) value of <0.05 was considered statistically significant.

## 3. Results

For the first time, the present study compared paired values of UACR, UPCR and the UACR-to-UPCR ratio in 47 children at early stages of CKD due to AS. In addition, the present study investigated variances in UACRs and UPCRs collected from spontaneous urine versus 24 h urine in 17 children. All patients were at early stages of AS and were examined and characterized very regularly, every 6 months, for years in the context of this ICH-GCP-conforming clinical study. The clear genetic diagnosis of AS and low comorbidities made these patients ideal candidates to compare the UACR and UPCR in each individual child over a long time period of up to 6 years. In addition, it is important to know that AS is—by genetic definition—a progressive disease that leads to kidney fibrosis. Ramipril has been shown to delay disease progression but has no potential to stop progression.

### 3.1. Baseline Characteristics

Table 1 shows albuminuria and proteinuria data from 47 individuals who participated in the EARLY PRO-TECT Alport trial; 45 (95.7%) patients were male. The mean age at sample-taking was 10.4 ± 4.5 years. All participants were at early stages of CKD with normal eGFRs (eGFR range of 107.6–151.5 mL/min at screening) [11]. Of forty-seven patients, thirty-eight (80.9%) had variants in the COL4A5 gene (X-linked inheritance), seven (15.9%) patients had variants in the COL4A3 or COL4A4 gene (autosomal recessive AS) and there was an unknown mode of inheritance in two patients (4.3%).

Of note is that all children had normal systolic and diastolic blood pressures according to their age-related percentiles. All children were nonsmokers. The patients’ median age was 8 years (interquartile range (IQR) of 8). The treatment phase lasted from 3 to up to 6 years.

### 3.2. Albuminuria, Proteinuria and Albuminuria-to-Proteinuria Ratios in All Participants

Collection of 24 h urine was successfully carried out in 21 children (all male). The mean age at collection was 11.6 (±4.7) years. All children except one (20/21, 95.2%) were treated with ACEi when the samples were taken. The mean UACR in 24 h urine was 332.6 (±460.8) mg/gCrea. The mean UPCR in 24 h urine was 560.0 (±529.6) mg/gCrea. Therefore, the mean albuminuria-to-proteinuria ratio in 24 h urine was 59.4%.

Spot-urine samples were collected from 46 children (two female, 4.3%) at a mean age of 10.5 (±4.5) years; 40 of 46 (86.9%) were treated with ACEi. The UACR in spot urine was 337.3 (±605.8) mg/g of creatinine, and the UPCR in spot urine was 551.2 (±724.1) mg/gCrea. Thus, the ratio of albuminuria to proteinuria in spot urine was 61.2% (similar to the ratio in 24 h urine, 59.4%), indicating that differences in proteinuria and albuminuria between spot urine and 24 h urine are minimal (Table 2).

### 3.3. Albuminuria, Proteinuria and Albuminuria-to-Proteinuria Ratios in Alport Stage 0, 1 and 2 Subgroups

The urine samples were divided further into three subgroups according to the current stage of AS: stage 0, albuminuria <30 mg/gCrea; stage 1, 30–300 mg/gCrea; and stage 2, albuminuria >300 mg/gCrea [11]. In these subgroups, rather significant differences between the UACR and the UPCR were identified depending on the stage and course of AS (Figure 1 and Figure 2).

In spot-urine samples from children in stage 0, the proteinuria was almost ten times higher than the albuminuria (106.4 ± 42.2 vs. 12.5 ± 9.7; *n* = 5; *p* < 0.05); it was “only” about three times higher in stage 1 (328.5 ± 210.1 vs. 132.3 ± 80.5; *n* = 9; *p* < 0.05) and almost equal in stage 2 (1481.9 ± 983.4 vs. 1109.7 ± 873.6; *n* = 7; *p* = 0.36) (Figure 1 and Table 1). These striking differences in the UACR and UPCR depending on the stage of AS in the spot urine were confirmed in the 24 h urine samples (Table 1, Figure 2).

### 3.4. Comparison of Albuminuria, Proteinuria and Albuminuria-to-Proteinuria Ratios in 24 h Urine vs. Spot Urine

The amount of albuminuria is a predictor of disease progression for most CKDs. In our study, however, at early stages of AS (stage 0 and 1), the amount of albuminuria reflected only a minority of all proteins in the urine (Figure 3) [2]. This visualization of the total range of proteins other than albumin allowed an objective impression of the sizes and total amount of proteins in the urine. The sizes of proteins ranged from about 30 to 170 kDa. Proteins smaller than albumin represented tubular damage or an overload of the tubular reabsorption capacity. Proteins larger than albumin represented glomerular damage.

In 17 patients from six different trial sites, UACRs and UPCRs were measured simultaneously in 24 h urine and spot urine in the same study visit (Table 2 and Figure 4). All children except one were treated with ACEi at the time when the samples were taken. Interestingly, albuminuria in 24 h urine vs. spot urine varied by less than 10% (266.8 ± 426.4 vs. 291.2 ± 530.2; *n* = 17) (Table 2). The same applies for proteinuria, again with a less than 10% difference between the 24 h urine and the spot urine (471.5 ± 504.9 vs. 480.2 ± 632.1; *n* = 17). When children with overt albuminuria higher than 300 mg/gCrea (Stage 2) were excluded, the albuminuria differed by less than 4% in the 24 h urine vs. the spot urine (120.4 ± 153.4 vs. 116 ± 120.3; *n* = 13) (Table 2 and Figure 4).

## 4. Discussion

The UACR and UPCR are key endpoints in most clinical trials assessing risk of progression of CKD. In the EARLY PRO-TECT Alport trial, for example, the UACR and UPCR were both measured in spot urine and in 24 h urine to monitor AS disease progression, which was the primary efficacy endpoint of the trial. While significant changes in a UACR are accepted by the FDA as surrogate parameters for CKD progression, the meanings of UACR-to-UPCR ratios in children at early stages of CKD and of variances in UACRs and UPCRs collected from spontaneous urine versus 24 h urine are less clear.

Here, we describe prominent differences in UACR to UPCR ratios in children with albuminuria <30 mg/gCrea (AS stage 0, ratio of about 1:10 UACR to UPCR) and with microalbuminuria (AS stage 1, ratio of about 1:3) or albuminuria >300 mg/gCrea (AS stage 2, ratio of 1:1.3). These findings were very similar in the spot urine compared to the 24 h urine obtained in the same study visit.

Previous studies have analyzed the relationship between the UACR and the UPCR in various kidney diseases, but not at very early stages of disease (prior onset of microalbuminuria) and not specifically for AS. The UACR/UPCR ratios varied in these studies depending on the amount of proteinuria, from 20% to 30% at low levels of proteinuria and rising to around 70% at higher levels [16,17]. Weaver et al. developed equations to estimate the median UACR from the median UPCR, optionally including specified covariates. For that, they used paired data from the same samples from a population-based cohort of more than 47,000 adults [18]. Consistently with prior research, they found substantial variation in the median urinary albumin-to-protein ratio, with the median ratio approaching 70% in severe proteinuria (>500 mg/gCrea) but dropping down to below 30% in less-severe proteinuria (<150 mg/gCrea) [18]. Our study showed quite similar results for overt proteinuria (UACR-to-UPCR ratio of 68.9% for UPCR > 500 mg/gCrea) and in patients at earlier stages of disease yet with milder amounts of proteinuria (UACR-to- PCR ratio of 16.3% for UPCR < 150 mg/gCrea) (Figure 1 and Figure 2). Therefore, the equation by Weaver et al. might also be used for patients with AS.

To our knowledge, no study ever has analyzed whether albuminuria or proteinuria is the better predictor for prognosis of AS or progression of disease. According to our results, future clinical trials should measure both albuminuria and proteinuria in parallel to examine which parameter better estimates a patient’s stage and prognosis of the kidney function. Since we found significantly higher proteinuria than albuminuria at earlier, preclinical stages of disease, it can be speculated that the UPCR and the UACR/UPCR ratio might be better early markers for kidney damage and development of CKD. According to our data, the filtration properties of the glomerular basement membrane and the capability to hold back larger and smaller proteins seem to change during the course of AS. The progressive loss of the filtration capacity of the glomerulus in AS might be caused by many different changes and impacts, which then together lead to progression of the disease: (1) altered assembly of the glomerular extracellular matrix; (2) changes in the (negative) charge of the basement membrane; (3) high (or low) matrix turnover; (4) faster degradation of defective type IV collagen proteins; (5) misfolding of proteins; (6) higher or lower expression of metalloproteinases (MMPs) and (7) tissue inhibitors of metalloproteinases (TIMPs); (8) progressive scar-tissue formation in the basement membrane; (9) impaired interaction with other components of the basement membrane, such as laminins, integrin or collagen receptors; and, finally, (10) cell apoptosis or loss of podocytes, with podocyte depletion leading to focal segmental glomerulosclerosis and massive loss of proteins, which are the characteristic histological features of AS in kidney biopsies in the later course of the disease [5,6]. The nephron-protective effect of Ramipril, however, seems to result from the fact that it favorably influences various steps in the AS pathogenesis [5,19].

Hereditary diseases of the glomerular filtration barrier, such as AS, are characterized by a more vulnerable glomerular basement membrane, leading to protein loss into the tubular system. Proteins including albumin can then be reabsorbed in the tubular system [20,21]. One can speculate that this system works better with albumin than with other proteins and works better with low (normal) amounts of albumin excretion but can soon be saturated by high albumin excretions. This might further damage the tubular system. This could explain why the proportion of albumin in proteinuria further increases during disease progression and why the UACR-to-UPCR ratio increased during AS progression in our study.

Progression of AS can be effectively delayed by an early, pre-emptive start of ACE inhibition. In individuals with less severe variants, ACE inhibition might even have the potential to delay renal failure for their lifetime [19]. To ensure this early therapy, early diagnosis is required. Further research is required to analyze nonalbumin proteins and identify valuable diagnostic biomarkers.

Our study had several limitations: the data about correlation of the amount of proteinuria with clinical outcomes in AS are sparse and mainly retrospective [10]. However, the primary efficacy endpoint in the first prospective trial for AS, EARLY PRO-TECT Alport, defined doubling or tripling of albuminuria as a progression marker, underscoring the great importance of proteinuria progression in the course of AS [11]. As a further limitation, the measurements of UACR and UPCR were carried out in 13 different laboratories. Other limitations of our study include the small sample size, the prominent male gender and the focus on Caucasian patients. In our study, 96% of the urine samples were taken from male patients with AS. Therefore, this study’s findings need to be validated in a larger sample size, and our results cannot be generalized. The clinical characteristics and the definitive genetic diagnoses of our young patients at early stages of kidney disease, however, partially compensate for these shortcomings. The strengths of our study include the ICH-GCP-conforming collection of data in a high-quality prospective double-blinded phase 3 clinical trial with very-well clinically and generically characterized children with AS. All families were advised to reduce their intakes of salt and animal-based food.

Many clinical trials use the UACR as a primary or a key secondary endpoint. Whether to use 24 h urine or spot urine, however, is less clear. Collecting urine for 24 h can be difficult in young children (and adults). Our data show that the differences in proteinuria and albuminuria between spot urine and 24 h urine are minimal (Table 2; Figure 4). Therefore, 24 h urine collection does not appear to be mandatory in children at early stages of CKD such as AS. This applies in particular to albuminuria levels below 300 mg/gCrea. In none of the children in our trial (*n* = 9), the difference between collection or spontaneous urine would have made a difference in the decision to start treatment with ACEi, according to the current recommendations [22]. Although the children participated in a clinical study and were thus repeatedly encouraged to collect 24 h urine, 24 h urine collection could be evaluated in less than 60% of the visits (351 of 613). This indicates—due to the young age of the children—that collecting urine is often not practical for children and their parents. Our findings have already been confirmed in other chronic kidney diseases [23,24]. Especially in small children (and, in our experience, in many adults as well), it is not realistic to expect correctly performed 24 h urine collection; our results indicate that spot urine is a reliable alternative for measuring the UACR and UPCR at early stages of CKD, particularly in patients with normal GFRs and albuminuria of less than 300 mg/gCrea.

## 5. Conclusions

Doubling or tripling of albuminuria was the primary efficacy endpoint of the EARLY PRO-TECT Alport trial. Any progress automatically led to unblinding. Therefore, the reliability and plausibility of measuring the albuminuria was crucial. As a consequence, we used a safety net of different collection methods to determine albuminuria. This allowed us, for the first time, to investigate the UACR, UPCR, UACR-to-UPCR ratio and relation of UACR/UPCR in spot urine compared to 24 h urine in patients at early stages of CKD (defined as a normal eGFR). Two important conclusions can be drawn from our study:(1)Both the UACR and the UPCR are relevant diagnostic biomarkers in patients with AS. As UPCRs and UACRs differed significantly in patients with AS and with yet low amounts of albuminuria, our study findings advocate for simultaneous measurements of the UPCR and the UACR in daily practice in early stages of AS. Use of the UPCR seems to be a key parameter at very early stages of glomerular diseases such as AS, noticeably earlier than the UACR. In general, the UACR and the UPCR are both recommended in daily clinical practice. At the very beginning of Alport syndrome, however, the UPCR (and the UACR-to-UPCR ratio) might be a very significant parameter to indicate how kidney function has changed over time.(2)At early stages of CKD with a normal GFR and a low amount of albuminuria, spot urine can be used as a reliable and convenient alternative to 24 h urine to monitor disease progression with the UACR and UPCR. The differences in proteinuria and albuminuria between spot urine and 24 h urine in our study were minimal. This finding has an important impact on future clinical trials in patients in early stages of CKD.

## Figures and Tables

**Figure 1 cells-12-01333-f001:**
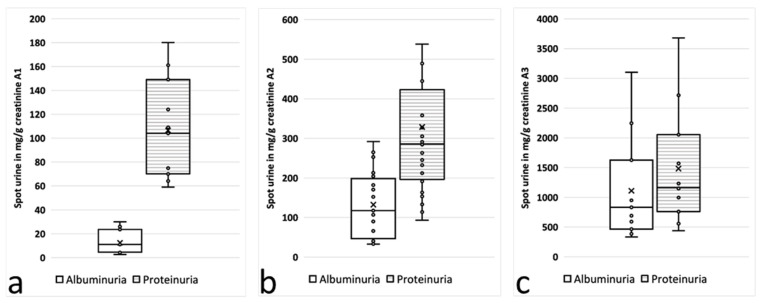
Albumin (plain) and protein (hatched) concentrations in spot urine: (**a**) AS stage 0, with albuminuria between 0 and <30 mg/gCrea at screening (*n* = 11); (**b**) AS stage 1, with albuminuria between 30 and 300 mg/gCrea at screening (*n* = 24); and (**c**) AS stage 2, with albuminuria >300 mg/gCrea at screening (*n* = 11).

**Figure 2 cells-12-01333-f002:**
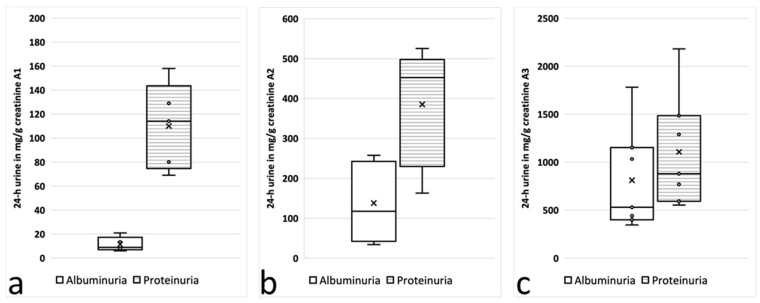
Albumin (plain) and protein (hatched) concentrations in 24 h urine collection: (**a**) AS stage 0, with albuminuria between 0 and <30 mg/gCrea at screening (*n* = 5); (**b**) AS stage 1, with albuminuria between 30 and 300 mg/gCrea at screening (*n* = 9); and (**c**) AS stage 2, with albuminuria >300 mg/gCrea at screening (*n* = 7).

**Figure 3 cells-12-01333-f003:**
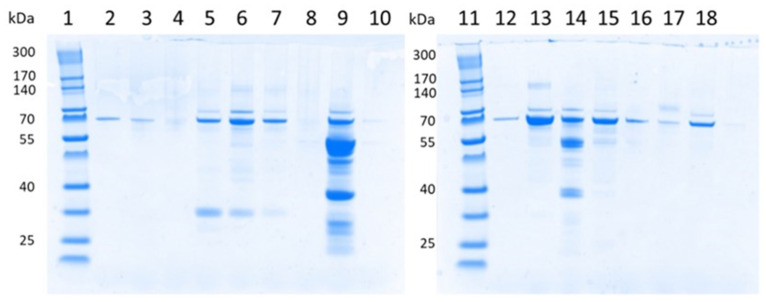
SDS-PAGE of urine samples from children with AS. Lane 1 + 11: molecular weight marker; lane 2 + 12: BSA-standard 300 ng; lane 3: patient 93-01-01, spot urine, visit 2 (month 12); lane 4: patient 93-01-02, 24 h urine, visit 4 (month 24); lane 5: patient 93-07-05, 24 h urine, visit 4 (month 24); lane 6: patient 93-07-08, 24 h urine, visit 7 (month 42); lane 7: patient 93-10-01, 24 h urine, visit 5 (month 30); lane 8: patient 93-10-02, 24 h urine, visit 6 (month 36); lane 9: patient 93-10-03, 24 h urine, visit 4 (month 24); lane 10: patient 93-11-03, spot urine, visit 3 (month 18); lane 12: patient 93-11-05, spot urine, visit 3 (month 18); lane 13: patient 93-12-05, 24 h urine, visit 4 (month 24); lane 14: patient 93-13-04, 24 h urine, visit 10 (month 60); lane 15: patient 93-14-02, 24 h urine, visit 3 (month 18); lane 16: patient 93-14-03, 24 h urine, visit 6 (month 36); lane 17: patient 93-16-01, spot urine, visit 3 (month 18); lane 18: patient 93-16-02, 24 h urine, visit 3 (month 18).

**Figure 4 cells-12-01333-f004:**
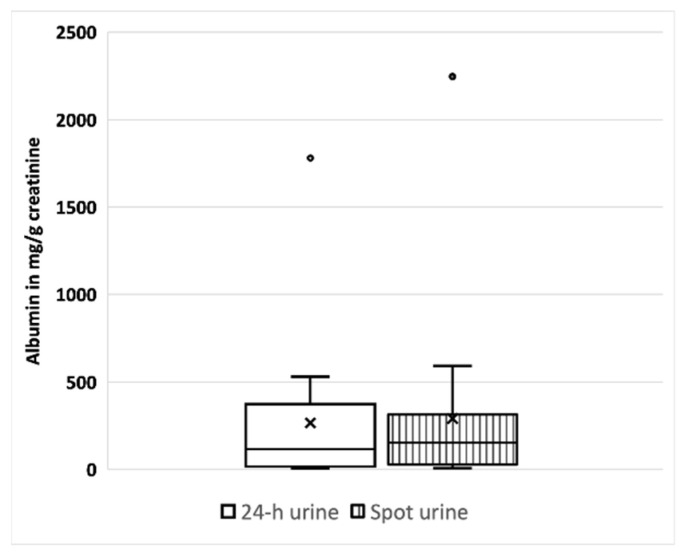
Albuminuria measured in 24 h urine (plain) and spot urine (striped) in the same study visit (*n* = 14).

**Table 1 cells-12-01333-t001:** Albuminuria, proteinuria, albuminuria-to-proteinuria ratios at different stages of AS.

Subgroup: Albuminuria (mg/gCrea)	Albuminuria in mg/gCrea (SD)	Proteinuria in mg/gCrea (SD)	Albuminuria/Proteinuria (%)	Age in Years (SD)	ACEi Therapy	*n*	*p*
** 24 h Urine **							
<30 (Alport stage 0)	11.4 (±6.0)	110 (±36.3)	10.4	12.2 (±3.6)	4/5 (80%)	5	<0.05
30–300 (Alport stage 1)	138 (±94.7)	385 (±140.8)	35.8	9.6 (±4.9)	9/9 (100%)	9	<0.05
>300 (Alport stage 2)	812.4 (±533.3)	1106.6 (±586.8)	73.4	13.7 (±4.7)	7/7 (100%)	7	0.35
** Spontaneous (Spot) Urine **							
<30 (Alport stage 0)	12.5 (±9.7)	106.4 (±42.2)	11.7	9.8 (±3.7)	9/11 (82%)	11	<0.05
30–300 (Alport stage 1)	132.3 (±80.5)	328.5 (±210.1)	40.3	10.2 (±4.9)	20/24 (83%)	24	<0.05
>300 (Alport stage 2)	1109.7 (±873.6)	1481.9 (±983.4)	74.9	11.7 (±4.6)	11/11 (100%)	11	0.36

**Table 2 cells-12-01333-t002:** Albuminuria and proteinuria measured in 24 h urine and spot urine in same study visit. First row: UACR and UPCR in mg/gCrea measured simultaneously; *n* = 17. Second row: UACR and UPCR in mg/gCrea, excluding children with albuminuria >300 mg/gCrea; *n* = 13.

Albuminuria 24 h Urine	Albuminuria Spot Urine	Proteinuria 24 h Urine	Proteinuria Spot Urine	Age (Years) at Collection
266.8 (±426.4)	291.2 (±530.2)	471.5 (±504.9)	480.2 (±632.1)	11.4 (±4.4)
120.4 (±153.4)	116 (±120.3)	279.6 (±187.1)	253.8 (±152.9)	11.5 (±4.7)

## Data Availability

Please send any additional requests to the corresponding author.

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
