# Peer review of "Ratio of Urinary Proteins to Albumin Excretion Shifts Substantially during Progression of the Podocytopathy Alport Syndrome, and Spot Urine Is a Reliable Method to Detect These Pathologic Changes"

_cells, 2023, doi:10.3390/cells12091333_

Round 1
Reviewer 1 Report
The authors showed measuring urinary albumin to creatinine ratio (UACR) and protein to creatinine ratio (UPCR) could be reliable diagnostic biomarkers in children with glomerular diseases. This report is interesting. However, there are some serious concerns in this manuscript described below. In my opinion, the significance of this report is not enough for the warrant publication in Cells.
Major points:
・I think this study is no randomized controlled trial about the comparison between UACR and UPCR. The authors should rethink and clarify the study design.
・I cannot fully understand the purpose of this research. It is unclear why the authors compared between UACR and UPCR or spot-urine and 24h-urine and why they measured albuminuria/proteinuria. The authors should clarify their hypothesis and purpose of this study. Especially, the intruduction should be simpler and clearer.
・What does the results of sodium dodecyl sulfate polyacrylamide gel electrophoresis of urine protein indicate?
Minor points:
・There are some errors in the description. Please correct them.
Author Response
Reviewer 1:
The authors showed measuring urinary albumin to creatinine ratio (UACR) and protein to creatinine ratio (UPCR) could be reliable diagnostic biomarkers in children with glomerular diseases. This report is interesting. However, there are some serious concerns in this manuscript described below. In my opinion, the significance of this report is not enough for the warrant publication in Cells.
Response: Thank you for the review and your critic, which we address in the response in order to further improve the text. We apology that there are some misunderstandings about the very relevant key message of our manuscript: “Spot urine is as good as 24-hour urine collection in order to assess UACR and UPCR, which both are key endpoints in most clinical trials assessing the risk of progression of chronic kidney disease (CKD)”.
This finding, for example, is critical for the ongoing clinical trial “FIONA” ClinicalTrial ID NCT05457283 (A Study to Learn More About How Safe the Study Treatment Finerenone is in Long-term Use When Taken With an ACE Inhibitor or Angiotensin Receptor Blocker Over 18 Months of Use in Children and Young Adults From 1 to 18 Years of Age With Chronic Kidney Disease and Proteinuria (FIONA OLE).
In addition, we – for the first time – describe changes in the UACR to UPCR ratio during disease progression of Alport-syndrome, which is, again, a very relevant finding for the future design of randomized controlled clinical trials.
As a response to the critic of reviewer 1, we clarified our findings throughout the text (see our response to your Major Points)
Major points:
・I think this study is no randomized controlled trial about the comparison between UACR and UPCR. The authors should rethink and clarify the study design.
Response: Sorry for this misunderstanding: the urine samples originate from the double-blinded randomized controlled trial EARLY PRO-TECT Alport with Ramipril versus Placebo in early stages of Alport syndrome. We did not randomize UACR vs. UPCR, but UACR was the key primary efficacy endpoint. Again, this indicates, how relevant our data are for future clinical trials.
As a response to the reviewer, we now clarified in the abstract (line 32 to 36) and discussion (lines 299-306) that we did not randomize for UACR/UPCR, but UACR/UPCR values originate from the randomized trial, which indicates the highest evidence level and high quality measures according to ICH-GCP principles.
Changes in the abstract (lines 32 to 36):
“For the first time, the current study compares UACR versus UPCR head-to-head at early stages of CKD, taking use of the hereditary podocytopathy Alport syndrome (AS) as model disease for any CKD. Urine samples originate from the prospective randomized controlled EARLY PRO-TECT Alport trial (NCT01485978). Urine samples from 47 children with a confirmed diagnosis of AS at very early stages of CKD were divided according to current stage of AS …”
Changes in the Discussion (lines 299 to 306):
“UACR and UPCR to creatinine ratio are key endpoints in most clinical trials assessing the risk of progression of CKD. In the EARLY PRO-TECT Alport trial, for example, UACR and UPCR both were measured in spot-urine and in 24-hour urine to monitor AS disease progression, which has been the primary efficacy endpoint of the trial. While significant changes in UACR are accepted by the FDA as surrogate parameter for CKD-progression, the meaning of UACR to UPCR ratio in children at early stages of CKD and of variances in UACR and UPCR collected from spontaneous urine versus 24-hour urine collection are less clear.”
・I cannot fully understand the purpose of this research. It is unclear why the authors compared between UACR and UPCR or spot-urine and 24h-urine and why they measured albuminuria/proteinuria. The authors should clarify their hypothesis and purpose of this study. Especially, the intruduction should be simpler and clearer.
Response: We understand your comment, however, many authors are involved in advisory/steering board meetings about clinical trials with CKD-patients. All meetings involve discussions about “how to assess UACR” (for example as repeated spot-urine measures on three consecutive days or as 24 hour urine etc.). Therefore, we address a very relevant question, which has not yet been addressed before in Alport syndrome, because EARLY PRO-TECT Alport has been the first randomized controlled trial.
As a response to the reviewer, we now clarify our hypothesis and purpose of this study in the introduction (lines 102 to 115).
Changes in the introduction (lines 102 to 115):
“For market approval of new nephroprotective drugs, the U.S. Food and Drug Administration (FDA) may accept relevant changes in UACR as primary endpoint as surrogate parameter for CKD-progression. Similarly, the phase 3 clinical trial EARLY PRO-TECT Alport measured proteinuria and albuminuria in order to assess the primary end-point in spontaneous urine and in a – if possible in the child – 24-hour urine collection every six months to monitor disease progression.
In the present study, we hypothesized that UACR to UPCR ratio in children with Alport syndrome at early stages of CKD changes during disease progression because of the increasing damage to the glomerular filtration unit. In addition, the purpose of this study is to evaluate variances in UACR and UPCR collected from spontaneous urine versus 24-hour urine collection in order to enrich statistical power calculations in future clinical trials aiming for market approval of new medications in patients at early stages of CKD.”
・What does the results of sodium dodecyl sulfate polyacrylamide gel electrophoresis of urine protein indicate?
Response: Thank you for this comment, which tells us that we need to explain the rationale for the SDS-gels better: At early stages of CKD, albumin represents only 10% of all proteins in the urine. Therefore, the key question is, what are the remaining 90%? The SDS-gel visualizes all proteins in the urine and is not limited to albumin. As shown in figure 3, size of proteins range from about 30 to 170 kDa. Proteins smaller than Albumin represent tubular damage or overload of small proteins (overload of tubular reabsorption capacity). Proteins larger than Albumin represent glomerular damage. We think it is very important to visualize these range of proteins to allow the reader to get an objective impression of size and amount of proteins in the urine.
As a response to the reviewer, we added this explanation in the result section in lines 270 to 274.
Changes in the discussion (lines 270 to 274):
“This visualization of the total range of proteins other than albumin allow an objective impression of size and amount of proteins in the urine. Size of proteins range from about 30 to 170 kDa. Proteins smaller than albumin represent tubular damage or over-load of tubular reabsorption capacity. Proteins larger than albumin represent glomerular damage.”
Minor points:
・There are some errors in the description. Please correct them.
Response: Typo errors (see below, response to reviewer 2) have been corrected.
Reviewer 2:
thank you for letting me evaluate this nice study.
Response: Thank you for your review, which helps us to further improve the text.
Some comments:
- the number of subjects is not fully clear throughout the paper, please revise and simplify.
Response: We now clarify the numbers throughout the paper. In addition, all tables and figure legends show the numbers included for analysis:
Line 203: “47” added to the text.
Line 205: “in 17 children” added to the text.
Lines 261 to 263: “n=5”; n=9”; “n=7” added to the text.
Line 288: “n=14” was added to the figure legend.
- the authors claim that protein excretion correlates with outcomes, but none is provided in the study, please clarify and add follow-up.
Response: Thank you for this comment, you raise a very interesting point: Indeed, the only known published data (European Alport registry, Gross et al, Kidney Int 2012 [10]) about the correlation of the amount of proteinuria with clinical outcome are retrospective. The only prospective data (ATHENA study, NCT02136862) are still unpublished. However, in the EARLY PRO-TECT Alport trial defined doubling or tripling of albuminuria as primary efficacy endpoint, which underscores the great importance of proteinuria progression in the course of Alport syndrome.
As a response to the reviewer, we included this point as an important limitation of our study in the discussion section in lines 368 to 373: “Our study has several limitations: the data about correlation of the amount of proteinuria with clinical outcome in AS are sparse and mainly retrospective [10]. However, the primary efficacy endpoint in the first prospective trial in AS, EARLY PRO-TECT Alport, defined doubling or tripling of albuminuria as progression mark-er, underscoring the great importance of proteinuria progression in the course of AS [11]. As a further limitation, …”
- please describe in the methods the way 24 hours urine collection was performed, and the same for spot urine. The instructions are critical for contextualization, and for interpretation of results.
Response: 24 hour urine and spot urine were collected using standard operating procedures (SOPs) and electronic case report forms (eCRFs) as part of the phase 3 clinical trial, EARLY PRO-TECT Alport, according to the published trial protocol. 24 hour urine usually was collected in “collection containers” provided by the trial site. Time of start and end of collection of 24 hour urine was documented as well as total amount of urine. Collection started with the second morning urine (first morning urine was discarded) and ended with the first morning urine at the next day. Spot urine as midstream urine was collected in the trial center using the second or third morning urine (depending on arrival at trial site). Next, urine analysis was performed at the laboratory of the local trial sites. Finally, results of urine analysis were documented in the eCRF. The eCRF automatic program checked the pseudonymised results of albuminuria for the primary efficacy endpoint (doubling or tripling of albuminuria). In addition, progress of disease for each trial participant was checked by the coordinating principal investigator every two to three months.
As response to the reviewer, we now further explain this process in the methods section in lines 156 to 167:
“Spot urine and 24 hour urine were collected using standard operating procedures (SOPs) and electronic case report forms (eCRFs) as part of the phase 3 clinical trial, according to the published trial protocol. Time of start and end of collection of 24 hour urine was documented as well as total amount of urine. Collection started with the second morning urine (first morning urine was discarded) and ended with the first morning urine at the next day. Spot urine as midstream urine was collected in the trial center using the second or third morning urine (depending on arrival at trial site). Urine analysis was performed at the laboratory of the local trial sites. The eCRF automatic program checked the results for plausibility (of creatinine levels and albumin levels in the urine) and for the primary efficacy endpoint (doubling or tripling of albuminuria). In addition, progress of disease for each trial participant was checked by the coordinating principal investigator every two to three months.”
- how did the authors control for the correct performance of 24 h urine collection? It is usually done based on coherence of creatininuria. The authors correctly underline the difficulties, so they should explain how they considered their collections reliable. 60% only of the required ones were delivered, but I cannot believe they were all reliable.
Response: Very good point. Indeed, we needed reliable results, because doubling or tripling of albuminuria was the primary efficacy endpoint in our phase 3 trial. The eCRF automatic program checked the results for plausibility (of creatinine levels and albumin levels in the urine) and checked the results of albuminuria for the primary efficacy endpoint (doubling or tripling of albuminuria). In addition, progress of disease for each trial participant was checked by the coordinating principal investigator every two to three months.” In addition, according to the trial protocol, we had a hierarchy for which sampling methods should be used to evaluate the primary efficacy endpoint: first, spot urine (because of a minimal risk for a sampling error or missing sample) and second, the 24 hour urine was used instead.
Of note, to our surprise, 60% delivery quote is similar or even better than delivery quotes in adult patients.
As a response to the reviewer, we included this information in the method sections in lines 156 to 167 (see above).
- Some clinical information could be of interest in the baseline table.
Response: Thank you for this comment. We added more clinical information in the “baseline characteristics” section in lines 221 to 224: “Of note, all children had a normal systolic and diastolic blood pressure according to their age-related percentiles. All children were non-smokers. The patients' median age was 8 years (interquartile range (IQR) 8). The treatment phase lasted from 3 to up to 6 years.”
- the first paragraph of the results is not a result and is a repetition, please erase.
Response: As a response to the other reviewer (see reviewer 1) and your comment, we replaced the redundant first paragraph of the discussion. The first paragraph (lines 299 to 306) now reads:
“UACR and UPCR to creatinine ratio are key endpoints in most clinical trials assessing the risk of progression of CKD. In the EARLY PRO-TECT Alport trial, for example, UACR and UPCR both were measured in spot-urine and in 24-hour urine to monitor AS disease progression, which has been the primary efficacy endpoint of the trial. While significant changes in UACR are accepted by the FDA as surrogate parameter for CKD-progression, the meaning of UACR to UPCR ratio in children at early stages of CKD and of variances in UACR and UPCR collected from spontaneous urine versus 24-hour urine collection are less clear.”
- the study is based on a small number of selected cases, this should be underlined
Response: Thank you, we mention the small number/ sample size as important limitation in the discussion (line 374). The cases, however, are not “selected” as they originate from a prospective clinical trial under ICH-GCP-conform conditions, which is an unique strength of our study.
- the authors correctly claim that 24 urine collections are often difficult and maybe unreliable, but they base their study on them... This should be the expanded.
Response: Thank you for your input. Indeed, we used a safety net of different collection methods to ensure reliability of the primary endpoint “amount of albuminuria” in our trial. We address this comment by the reviewer in lines 403 to 407:
“Doubling or tripling of albuminuria has been the primary efficacy endpoint of EARLY PRO-TECT Alport trial. Any progress automatically led to unblinding. Therefore, reliability and plausibility of measuring albuminuria was crucial. As a consequence, we used a safety net of different collection methods to determine albuminuria. This allows us, …”
- the discussion is a bit redundant, could be simplified made more crispy and critical.
Response: We understand your critic and like the “crispy”, however, two points caused this “redundance”: First, prior submission, the editors asked us to expand the discussion section by discussing the possible mechanisms (lines 332 to 353). Second, we have 19 authors, many of them contributed their part to the discussion (and they would give me a hard time to discard some of their changes and comments).
- some typos (none smokers for ex.)
Response: Thank you, I rechecked the text for typos.
Thank you very much. Kind regards, Oliver Gross and Jan Boeckhaus
Reviewer 2 Report
thank you for letting me evaluate this nice study.
Some comments:
- the number of subjects is not fully clear throughout the paper, please revise and simplify.
- the authors claim that protein excretion correlates with outcomes, but none is provided in the study, please clarify and add follow-up.
- please describe in the methods the way 24 hours urine collection was performed, and the same for spot urine. The instructions are critical for contextualization, and for interpretation of results.
- how did the authors control for the correct performance of 24 h urine collection? It is usually done based on coherence of creatininuria. The authors correctly underline the difficulties, so they should explain how they considered their collections reliable. 60% only of the required ones were delivered, but I cannot believe they were all reliable.
- Some clinical information could be of interest in the baseline table.
- the first paragraph of the results is not a result and is a repetition, please erase.
- the study is based on a small number of selected cases, this should be underlined
- the authors correctly claim that 24 urine collections are often difficult and maybe unreliable, but they base their study on them... This should be the expanded.
- the discussion is a bit redundant, could be simplified made more crispy and critical.
- some typos (none smokers for ex.)
Author Response
Reviewer 2:
thank you for letting me evaluate this nice study.
Response: Thank you for your review, which helps us to further improve the text.
Some comments:
- the number of subjects is not fully clear throughout the paper, please revise and simplify.
Response: We now clarify the numbers throughout the paper. In addition, all tables and figure legends show the numbers included for analysis:
Line 203: “47” added to the text.
Line 205: “in 17 children” added to the text.
Lines 261 to 263: “n=5”; n=9”; “n=7” added to the text.
Line 288: “n=14” was added to the figure legend.
- the authors claim that protein excretion correlates with outcomes, but none is provided in the study, please clarify and add follow-up.
Response: Thank you for this comment, you raise a very interesting point: Indeed, the only known published data (European Alport registry, Gross et al, Kidney Int 2012 [10]) about the correlation of the amount of proteinuria with clinical outcome are retrospective. The only prospective data (ATHENA study, NCT02136862) are still unpublished. However, in the EARLY PRO-TECT Alport trial defined doubling or tripling of albuminuria as primary efficacy endpoint, which underscores the great importance of proteinuria progression in the course of Alport syndrome.
As a response to the reviewer, we included this point as an important limitation of our study in the discussion section in lines 368 to 373: “Our study has several limitations: the data about correlation of the amount of proteinuria with clinical outcome in AS are sparse and mainly retrospective [10]. However, the primary efficacy endpoint in the first prospective trial in AS, EARLY PRO-TECT Alport, defined doubling or tripling of albuminuria as progression mark-er, underscoring the great importance of proteinuria progression in the course of AS [11]. As a further limitation, …”
- please describe in the methods the way 24 hours urine collection was performed, and the same for spot urine. The instructions are critical for contextualization, and for interpretation of results.
Response: 24 hour urine and spot urine were collected using standard operating procedures (SOPs) and electronic case report forms (eCRFs) as part of the phase 3 clinical trial, EARLY PRO-TECT Alport, according to the published trial protocol. 24 hour urine usually was collected in “collection containers” provided by the trial site. Time of start and end of collection of 24 hour urine was documented as well as total amount of urine. Collection started with the second morning urine (first morning urine was discarded) and ended with the first morning urine at the next day. Spot urine as midstream urine was collected in the trial center using the second or third morning urine (depending on arrival at trial site). Next, urine analysis was performed at the laboratory of the local trial sites. Finally, results of urine analysis were documented in the eCRF. The eCRF automatic program checked the pseudonymised results of albuminuria for the primary efficacy endpoint (doubling or tripling of albuminuria). In addition, progress of disease for each trial participant was checked by the coordinating principal investigator every two to three months.
As response to the reviewer, we now further explain this process in the methods section in lines 156 to 167:
“Spot urine and 24 hour urine were collected using standard operating procedures (SOPs) and electronic case report forms (eCRFs) as part of the phase 3 clinical trial, according to the published trial protocol. Time of start and end of collection of 24 hour urine was documented as well as total amount of urine. Collection started with the second morning urine (first morning urine was discarded) and ended with the first morning urine at the next day. Spot urine as midstream urine was collected in the trial center using the second or third morning urine (depending on arrival at trial site). Urine analysis was performed at the laboratory of the local trial sites. The eCRF automatic program checked the results for plausibility (of creatinine levels and albumin levels in the urine) and for the primary efficacy endpoint (doubling or tripling of albuminuria). In addition, progress of disease for each trial participant was checked by the coordinating principal investigator every two to three months.”
- how did the authors control for the correct performance of 24 h urine collection? It is usually done based on coherence of creatininuria. The authors correctly underline the difficulties, so they should explain how they considered their collections reliable. 60% only of the required ones were delivered, but I cannot believe they were all reliable.
Response: Very good point. Indeed, we needed reliable results, because doubling or tripling of albuminuria was the primary efficacy endpoint in our phase 3 trial. The eCRF automatic program checked the results for plausibility (of creatinine levels and albumin levels in the urine) and checked the results of albuminuria for the primary efficacy endpoint (doubling or tripling of albuminuria). In addition, progress of disease for each trial participant was checked by the coordinating principal investigator every two to three months.” In addition, according to the trial protocol, we had a hierarchy for which sampling methods should be used to evaluate the primary efficacy endpoint: first, spot urine (because of a minimal risk for a sampling error or missing sample) and second, the 24 hour urine was used instead.
Of note, to our surprise, 60% delivery quote is similar or even better than delivery quotes in adult patients.
As a response to the reviewer, we included this information in the method sections in lines 156 to 167 (see above).
- Some clinical information could be of interest in the baseline table.
Response: Thank you for this comment. We added more clinical information in the “baseline characteristics” section in lines 221 to 224: “Of note, all children had a normal systolic and diastolic blood pressure according to their age-related percentiles. All children were non-smokers. The patients' median age was 8 years (interquartile range (IQR) 8). The treatment phase lasted from 3 to up to 6 years.”
- the first paragraph of the results is not a result and is a repetition, please erase.
Response: As a response to the other reviewer (see reviewer 1) and your comment, we replaced the redundant first paragraph of the discussion. The first paragraph (lines 299 to 306) now reads:
“UACR and UPCR to creatinine ratio are key endpoints in most clinical trials assessing the risk of progression of CKD. In the EARLY PRO-TECT Alport trial, for example, UACR and UPCR both were measured in spot-urine and in 24-hour urine to monitor AS disease progression, which has been the primary efficacy endpoint of the trial. While significant changes in UACR are accepted by the FDA as surrogate parameter for CKD-progression, the meaning of UACR to UPCR ratio in children at early stages of CKD and of variances in UACR and UPCR collected from spontaneous urine versus 24-hour urine collection are less clear.”
- the study is based on a small number of selected cases, this should be underlined
Response: Thank you, we mention the small number/ sample size as important limitation in the discussion (line 374). The cases, however, are not “selected” as they originate from a prospective clinical trial under ICH-GCP-conform conditions, which is an unique strength of our study.
- the authors correctly claim that 24 urine collections are often difficult and maybe unreliable, but they base their study on them... This should be the expanded.
Response: Thank you for your input. Indeed, we used a safety net of different collection methods to ensure reliability of the primary endpoint “amount of albuminuria” in our trial. We address this comment by the reviewer in lines 403 to 407:
“Doubling or tripling of albuminuria has been the primary efficacy endpoint of EARLY PRO-TECT Alport trial. Any progress automatically led to unblinding. Therefore, reliability and plausibility of measuring albuminuria was crucial. As a consequence, we used a safety net of different collection methods to determine albuminuria. This allows us, …”
- the discussion is a bit redundant, could be simplified made more crispy and critical.
Response: We understand your critic and like the “crispy”, however, two points caused this “redundance”: First, prior submission, the editors asked us to expand the discussion section by discussing the possible mechanisms (lines 332 to 353). Second, we have 19 authors, many of them contributed their part to the discussion (and they would give me a hard time to discard some of their changes and comments).
- some typos (none smokers for ex.)
Response: Thank you, I rechecked the text for typos.
Thank you very much. Kind regards, Oliver Gross and Jan Boeckhaus
Reviewer 1:
The authors showed measuring urinary albumin to creatinine ratio (UACR) and protein to creatinine ratio (UPCR) could be reliable diagnostic biomarkers in children with glomerular diseases. This report is interesting. However, there are some serious concerns in this manuscript described below. In my opinion, the significance of this report is not enough for the warrant publication in Cells.
Response: Thank you for the review and your critic, which we address in the response in order to further improve the text. We apology that there are some misunderstandings about the very relevant key message of our manuscript: “Spot urine is as good as 24-hour urine collection in order to assess UACR and UPCR, which both are key endpoints in most clinical trials assessing the risk of progression of chronic kidney disease (CKD)”.
This finding, for example, is critical for the ongoing clinical trial “FIONA” ClinicalTrial ID NCT05457283 (A Study to Learn More About How Safe the Study Treatment Finerenone is in Long-term Use When Taken With an ACE Inhibitor or Angiotensin Receptor Blocker Over 18 Months of Use in Children and Young Adults From 1 to 18 Years of Age With Chronic Kidney Disease and Proteinuria (FIONA OLE).
In addition, we – for the first time – describe changes in the UACR to UPCR ratio during disease progression of Alport-syndrome, which is, again, a very relevant finding for the future design of randomized controlled clinical trials.
As a response to the critic of reviewer 1, we clarified our findings throughout the text (see our response to your Major Points)
Major points:
・I think this study is no randomized controlled trial about the comparison between UACR and UPCR. The authors should rethink and clarify the study design.
Response: Sorry for this misunderstanding: the urine samples originate from the double-blinded randomized controlled trial EARLY PRO-TECT Alport with Ramipril versus Placebo in early stages of Alport syndrome. We did not randomize UACR vs. UPCR, but UACR was the key primary efficacy endpoint. Again, this indicates, how relevant our data are for future clinical trials.
As a response to the reviewer, we now clarified in the abstract (line 32 to 36) and discussion (lines 299-306) that we did not randomize for UACR/UPCR, but UACR/UPCR values originate from the randomized trial, which indicates the highest evidence level and high quality measures according to ICH-GCP principles.
Changes in the abstract (lines 32 to 36):
“For the first time, the current study compares UACR versus UPCR head-to-head at early stages of CKD, taking use of the hereditary podocytopathy Alport syndrome (AS) as model disease for any CKD. Urine samples originate from the prospective randomized controlled EARLY PRO-TECT Alport trial (NCT01485978). Urine samples from 47 children with a confirmed diagnosis of AS at very early stages of CKD were divided according to current stage of AS …”
Changes in the Discussion (lines 299 to 306):
“UACR and UPCR to creatinine ratio are key endpoints in most clinical trials assessing the risk of progression of CKD. In the EARLY PRO-TECT Alport trial, for example, UACR and UPCR both were measured in spot-urine and in 24-hour urine to monitor AS disease progression, which has been the primary efficacy endpoint of the trial. While significant changes in UACR are accepted by the FDA as surrogate parameter for CKD-progression, the meaning of UACR to UPCR ratio in children at early stages of CKD and of variances in UACR and UPCR collected from spontaneous urine versus 24-hour urine collection are less clear.”
・I cannot fully understand the purpose of this research. It is unclear why the authors compared between UACR and UPCR or spot-urine and 24h-urine and why they measured albuminuria/proteinuria. The authors should clarify their hypothesis and purpose of this study. Especially, the intruduction should be simpler and clearer.
Response: We understand your comment, however, many authors are involved in advisory/steering board meetings about clinical trials with CKD-patients. All meetings involve discussions about “how to assess UACR” (for example as repeated spot-urine measures on three consecutive days or as 24 hour urine etc.). Therefore, we address a very relevant question, which has not yet been addressed before in Alport syndrome, because EARLY PRO-TECT Alport has been the first randomized controlled trial.
As a response to the reviewer, we now clarify our hypothesis and purpose of this study in the introduction (lines 102 to 115).
Changes in the introduction (lines 102 to 115):
“For market approval of new nephroprotective drugs, the U.S. Food and Drug Administration (FDA) may accept relevant changes in UACR as primary endpoint as surrogate parameter for CKD-progression. Similarly, the phase 3 clinical trial EARLY PRO-TECT Alport measured proteinuria and albuminuria in order to assess the primary end-point in spontaneous urine and in a – if possible in the child – 24-hour urine collection every six months to monitor disease progression.
In the present study, we hypothesized that UACR to UPCR ratio in children with Alport syndrome at early stages of CKD changes during disease progression because of the increasing damage to the glomerular filtration unit. In addition, the purpose of this study is to evaluate variances in UACR and UPCR collected from spontaneous urine versus 24-hour urine collection in order to enrich statistical power calculations in future clinical trials aiming for market approval of new medications in patients at early stages of CKD.”
・What does the results of sodium dodecyl sulfate polyacrylamide gel electrophoresis of urine protein indicate?
Response: Thank you for this comment, which tells us that we need to explain the rationale for the SDS-gels better: At early stages of CKD, albumin represents only 10% of all proteins in the urine. Therefore, the key question is, what are the remaining 90%? The SDS-gel visualizes all proteins in the urine and is not limited to albumin. As shown in figure 3, size of proteins range from about 30 to 170 kDa. Proteins smaller than Albumin represent tubular damage or overload of small proteins (overload of tubular reabsorption capacity). Proteins larger than Albumin represent glomerular damage. We think it is very important to visualize these range of proteins to allow the reader to get an objective impression of size and amount of proteins in the urine.
As a response to the reviewer, we added this explanation in the result section in lines 270 to 274.
Changes in the discussion (lines 270 to 274):
“This visualization of the total range of proteins other than albumin allow an objective impression of size and amount of proteins in the urine. Size of proteins range from about 30 to 170 kDa. Proteins smaller than albumin represent tubular damage or over-load of tubular reabsorption capacity. Proteins larger than albumin represent glomerular damage.”
Minor points:
・There are some errors in the description. Please correct them.
Response: Typo errors (see below, response to reviewer 2) have been corrected.
Round 2
Reviewer 1 Report
I would like to thank you for your revise. I can understand your answers for my questions. Finally, I think that this manuscript is acceptable for the warrant publication in Cells.